# Dried Herbs as an Easy-to-Use and Cost-Effective Alternative to Essential Oils to Extend the Shelf Life of Sheep Lump Cheese

**DOI:** 10.3390/foods12244487

**Published:** 2023-12-15

**Authors:** Simona Kunová, Isabella Taglieri, Peter Haščík, Anis Ben Hsouna, Wissem Mnif, Francesca Venturi, Chiara Sanmartin, Natália Čmiková, Maciej Ireneusz Kluz, Miroslava Kačániová

**Affiliations:** 1Institute of Food Technology, Faculty of Biotechnology and Food Sciences, Slovak University of Agriculture, Trieda A. Hlinku 2, 94976 Nitra, Slovakiapeter.hascik@uniag.sk (P.H.); 2Department of Agriculture, Food and Environment, University of Pisa, Via del Borghetto 80, 56126 Pisa, Italychiara.sanmartin@unipi.it (C.S.); 3Nutrafood Research Center, University of Pisa, Via del Borghetto 80, 56124 Pisa, Italy; 4Laboratory of Biotechnology and Plant Improvement, Centre of Biotechnology of Sfax, Sfax 3038, Tunisia; benhsounanis@yahoo.fr; 5Department of Environmental Sciences and Nutrition, Higher Institute of Applied Sciences and Technology of Mahdia, University of Monastir, Monastir 5000, Tunisia; 6Department of Chemistry, College of Sciences at Bisha, University of Bisha, P.O. Box 199, Bisha 61922, Saudi Arabia; 7Institute of Horticulture, Faculty of Horticulture and Landscape Engineering, Slovak University of Agriculture, Tr. A. Hlinku 2, 94976 Nitra, Slovakia; 8School of Medical & Health Sciences, University of Economics and Human Sciences in Warsaw, Okopowa 59, 01 043 Warszawa, Poland; m.kluz@vizja.pl

**Keywords:** ewes cheese, sheep lump cheese, essential oils, dried herbs, microbiological quality, identification of microorganisms

## Abstract

The objective of this research was to assess the effectiveness of three specific dried herbs (rosemary, thyme, and oregano) in combating microbial spoilage in sheep lump cheese. This was achieved by comparing them with a control group and cheeses treated with corresponding 1% essential oils (*Rosmarinus officinalis*, *Origanum vulgare*, *Thymus vulgaris*). All cheese samples were vacuum-sealed and stored at 4 °C for 15 days. Analysis of total viable counts of viable bacteria (TVC), coliform bacteria (CB), lactic acid bacteria (LAB), and microscopic filamentous fungi (MFF) was conducted on days 0, 5, 10, and 15. The results revealed that, at the end of the storage period, dried oregano-treated samples exhibited the lowest TVC count (5.80 log CFU/g), while dried rosemary-treated samples showed the lowest CB count (3.27 log CFU/g). Moreover, the lowest MFF count (2.40 log CFU/g) was observed in oregano essential oil-treated samples. Additionally, dried oregano-treated samples displayed the highest LAB count (4.49 log CFU/g) at the experiment’s conclusion. Furthermore, microorganism identification from sheep cheese was performed using MALDI-TOF MS Biotyper technology, revealing that the most frequently isolated bacteria were *Citrobacter braakii* and *Hafnia alvei* (Enterobacteriaceae family), along with *Lacticaseibacillus paracasei* (Lactobacillaceae family). In summary, all the natural substances examined exhibited inhibitory effects against the studied microorganisms, with oregano essential oil and dried oregano demonstrating the strongest inhibitory effects. This supports their potential use as cost-effective natural preservatives to extend the shelf life of sheep lump cheese.

## 1. Introduction

In light of substantial global food production and the ongoing issue of widespread food wastage [1], the preservation of food emerges as a crucial factor in ensuring food security by averting potential contamination and simultaneously addressing the current problem of waste [2]. Within the context of the Mediterranean diet, dairy products contribute vital nutrients, including certain short-chain and trans-fatty acids, high-quality proteins, as well as essential vitamins and minerals [3,4]. Cheese, within the spectrum of dairy products, serves as a compelling economic resource in marginalized regions, even in countries characterized by hot climates [5]. Generally, cheese is considered a safe food, with susceptibility to microbial spoilage limited to specific types. This susceptibility is influenced by various factors, including the cheese’s characteristics such as chemical composition, pH, water content, temperature, and processing conditions [6]. Moreover, the swift globalization of the cheese supply chain has increased the likelihood of contamination at various stages of the production process, leading to occasional recalls and outbreaks [7]. This trend underscores the need for innovative approaches to cheese preservation

Various microorganism groups, including potential foodborne pathogens, may be found in milk and its derivatives. The primary sources of pathogens in milk typically include the farm environment, the mammary gland, or animals with diseases [8]. Among the most frequently encountered foodborne pathogens originating from milk are *Brucella* spp., *Staphylococcus aureus*, *Escherichia coli*, and *Salmonella* spp. [9]. Yeasts and molds also contribute significantly to the spoilage of dairy products, potentially leading to alterations in taste, texture, and color in cheeses. Furthermore, the proliferation of molds in cheese has the potential to pose health risks to consumers [10].

In recent decades, traditional saturated Generally Recognized as Safe (GRAS) substances and/or conventional food additives that were commonly utilized for preserving cheese, such as BHT, BHA, sulfites, propionates, sorbates, acetic acid, hexamethylene tetramines, and benzoates, have seen a growing shift towards natural preservatives [11]. This transition aligns with consumer preferences for healthier food options and “clean labels” [12]. Essential oils (EOs) and their constituent compounds have become a focal point in the exploration of natural preservatives due to their documented antibacterial, antifungal, antiviral, and insecticidal properties Notably, rosemary essential oils have demonstrated the ability to inhibit a broad spectrum of bacteria, including *Listeria monocytogenes, Escherichia coli*, *Pseudomonas fluorescens*, *Bacillus cereus*, *Staphylococcus aureus*, and *Latilactobacillus sakei* when incorporated into certain food products [13,14,15].

The essential oil derived from *Thymus vulgaris* leaves has the potential for use as an aromatic spice additive in various products, including food items, cosmetics, and pharmaceuticals. Notably, thymol and carvacrol, prominent constituents of thyme essential oil, exhibit extensive antimicrobial and antioxidant properties [16]. Specifically, thyme essential oil demonstrates efficacy against *Aspergillus flavus*, *Escherichia coli*, *Staphylococcus aureus*, and *Listeria monocytogenes* [17]. Oregano essential oil stands out as one of the most potent antimicrobial and antioxidant agents, with even low concentrations showing a broad spectrum of activity against both pathogenic and spoilage bacteria [18]. Numerous studies have highlighted the effectiveness of oregano essential oil as an antioxidant and flavouring agent in functional food or nutraceutical products [19].

On the flip side, the cultivation of lactic acid bacteria (LAB) is typically essential to facilitate the maturation of cheese. Numerous studies [20,21,22,23,24] have noted that LAB found in cheese are generally resilient to the inhibitory concentrations of plant extracts and essential oils (EOs) that affect the growth of pathogenic bacteria. Despite the proven efficacy of most essential oils in preserving cheese, several factors may constrain their widespread use. Challenges include the potential for intense flavor and interactions with food ingredients, limiting their applicability. Additionally, considerations such as cost [25,26,27] and potential toxicity at high doses, despite their Generally Recognized as Safe (GRAS) status, pose further obstacles. To address these limitations, the literature has also highlighted the effectiveness of spices and dried herbs as natural antimicrobial compounds [28,29]. Spices and herbs contain aromatic and volatile compounds known for their antimicrobial properties [26], which are present in various plant parts such as flowers, roots, and bark, serving as natural defense mechanisms against predators [30]. Furthermore, these plant compounds possess antioxidant properties and contribute to enhancing color and flavor [31]. Plants *Rosmarinus officinalis*, *Origanum vulgare*, and *Thymus vulgaris* belong to the Lamiaceae family, of which several species show high pesticidal activity, and are prospective bioagents against several detrimental pests [32,33]. Rosemary, recognized as a medicinal plant, boasts a high concentration of aromatic phenolic compounds with nutraceutical and pharmaceutical benefits, including anti-obesity, anti-inflammatory, antidiabetic, diuretic, antithrombotic, antimicrobial, anticancer, hepatoprotective, and antioxidant properties. Due to its capacity to prevent oxidation and microbial contamination, rosemary extract has been employed in food preservation, offering a potential alternative to or reduction in synthetic antioxidants in food products [34].

Thyme is utilized both in its fresh form and as dried leaves, playing a pivotal role as the most significant medicinal plant consumed across diverse regions worldwide [35]. Various research studies have demonstrated that incorporating thyme into foods (such as meat, meat products, milk, fish, or fish products) enhances stability and reduces lipid oxidation during the shelf-life period. This positions thyme as a promising natural additive source [34]. Oregano finds applications in a range of culinary contexts, including meat, sausages, salads, stews, dressings, and soups. In the food industry, both oregano oil and oregano resin are employed in the production of foods and beverages, and oregano is also featured in cosmetic formulations. Oregano oil, specifically, is utilized in alcoholic beverages, baked goods, meats, condiments, relishes, milk products, processed vegetables, snack foods, fats, and oils. Notably, it is a prevalent spice in pizza recipes and, alongside black pepper, serves as a common ingredient in dressings, acting as a viable substitute for table salt. Ground oregano, when present at a concentration of 2%, has demonstrated robust antifungal properties against several molds that commonly contaminate food [36]. Although the use of herbs and spices in cheese production dates back to ancient times [37], research focusing on the effectiveness of dried herbs in preserving cheese from microbial spoilage remains relatively limited in the existing literature.

In this study, the primary objective was to assess the antimicrobial impact of three specific dried herbs and their corresponding essential oils, namely rosemary (*Rosmarinus officinalis*,), thyme (*Thymus vulgaris*), and oregano (*Origanum vulgare*)), from the family Lamiaceae. The evaluation focused on their efficacy against microbial spoilage in sheep lump cheese, which was stored under vacuum conditions for a duration of 15 days at a temperature of 4 °C.

## 2. Materials and Methods

### 2.1. Preparation and Packaging of Ewes Cheese Samples

For this experiment, Ewe’s cheese was utilized and procured from a private seller in Turčianske Teplice, Slovakia (48°51′32″ N 18°51′49″ E). The physico-chemical characteristics of the raw ewe’s cheese were as follows: pH 5.1, salt content 2.4%, protein content 24%, moisture content 37%, and fat content 31%. A total of 4000 g of cheese was obtained and transported to the microbiological laboratory under refrigeration conditions. The ewe’s cheese was allowed to drain at 18 °C. Subsequently, approximately 25 g portions of the ewe’s cheese were cut with a sterile knife and individually weighed.

Each 25 g cheese sample, excluding the control group, was treated with 100 µL of 1% *Rosmarinus officinalis* EO (REO), 1% *Thymus vulgaris* EO (TEO), and 1% *Origanum vulgare* EO (OEO) obtained from Hanus Ltd., Nitra, Slovakia. Additionally, dried herbs (rosemary—DR, thyme—DT, oregano—DO) at a concentration of 1% were applied.

*Rosmarinus officinalis* EO was produced through steam distillation of flowering clematis, with main components including 1,8 cineol (38–55%), camphor (5–15%), α+β-pinene (13–23%), limonene (1–4%), and borneol (1–5%). *Origanum vulgare* EO, obtained through steam distillation of fresh wort, consisted predominantly of carvacrol (70.8%), thymol (3.9%), p-cymene (5.6%), γ-terpinene (4.8%), β-caryophyllene (2.16%), α-terpinene (1.06%), β-myrcene, and α-pinene. *Thymus vulgaris* EO, produced by steam distillation of partially dried extract, had major constituents such as thymol (36–55%), p-cymene (15–28%), carvacrol (1–4%), γ-terpinene (5–10%), and linalool (4–6%) as reported by the manufacturer. The dried herbs (*Rosmarinus officinalis*, *Origanum vulgare*, *Thymus vulgaris*) were purchased from Mäspoma Ltd. (Zvolen, Slovakia) and were microbiologically tested prior to application on cheese samples, with no bacterial presence detected.

Dried herbs and sunflower oil were microbiologically tested before their application to cheese samples. Microbiological analysis of herbs and oil were conducted without bacteria. Essential oils were dissolved in sunflower oil and dried herbs were mixed with oil. One gram of herbs was mixed with 100 mL of sunflower oil. The samples of ewe’s cheese were soaked with 1% EO and oil mixed with herbs (1%) for 1 h. The samples were packed in polyethylene bags using a vacuum packer (Concept, Choceň, Czech Republic). Each sample of cheese (25 g) was packaged individually. The groups of control samples (CA) were not vacuum packed, control samples were packed in vacuum (CV), and control samples with oil were vacuum packed. The samples with addition of herbs and EO (REO, TEO, OEO, DR, DT, DO) were prepared, and cheese samples were put into the main bags and mixed gently in the bag for 1 min approximately, so as not to damage the cheese sample, with the addition of 1% EO and herbs solution. After this operation they were vacuum packed (Table 1).

### 2.2. Samples Cultivation

Microbiological assessments were conducted on the 0th, 5th, 10th, and 15th days of storage at 4 °C. A quantity of 25 g of cheese samples was weighed and thereafter placed in an aseptic stomacher bag. After that, 225 mL of peptone water was used to dilute the samples to 10^−1^, and a Stomacher was used to homogenize them for two minutes. A 0.1 mL of an aliquot from appropriate dilution was pipetted and spread on standard pre-dried plate count agar media.

The homogenized samples underwent shaking in a shaker (GFL 3031, Burgwedel, Germany) for 30 min. Total viable counts (TVC) were determined using plate count agar (PCA, Oxoid, Basingstoke, UK) [38], and inoculated Petri dishes were incubated at 30 °C for 48–72 h. For the determination of coliform bacteria (CB), Violet Red Bile lactose agar (VRBL, Oxoid, Basingstoke, UK) [39] was utilized, and inoculated Petri dishes were incubated at 37 °C for 24–48 h. Lactic acid bacteria (LAB) [40] were enumerated using Rogosa and Sharpe agar (MRS, Oxoid, Basingstoke, UK), with inoculated Petri dishes incubated with 5% CO_2_ at 30 °C for 48–72 h. Microscopic filamentous fungi (MFF) were assessed using Dichloran-rose Bengal chloramphenicol agar (DRBC, Oxoid, Basingstoke, UK) [41], and inoculated Petri dishes were incubated at 25 °C for 5–7 days. All measurement analyses were conducted in triplicate.

### 2.3. Sample Preparation and MALDI-TOF MS Measurement

The identification of microorganisms isolated from ewe’s cheese samples was conducted using MALDI-TOF (matrix-assisted laser desorption/ionization time of flight) MS Biotyper from Bruker (Daltonics, Bremen, Germany). Bacterial and yeast colonies were subcultured on Tryptone Soya Agar (TSA agar, Oxoid, UK) for 18–24 h prior to identification. One colony from each of the eight bacterial isolates was selected. Bacterial and yeast colonies were suspended in a mixture of 300 μL distilled water (Sigma-Aldrich, St. Louis, MO, USA) and 900 μL absolute ethanol (Bruker Daltonik, Bremen, Germany), then centrifuged at 13,000 rpm for 2 min. The resulting pellet was mixed with 50 μL of 70% formic acid (*v*/*v*) (Sigma-Aldrich, USA) and 50 μL of acetonitrile (Sigma-Aldrich, USA) after removing the supernatant.

Following another centrifugation, 1 μL of the supernatant was applied to a steel plate and air-dried at 20 °C. Subsequently, 1 μL of MALDI matrix was applied to the samples. Mass spectra results were analyzed using the MALDI Biotyper 3.0 software (Bruker Daltonik, Germany). Identification criteria included a score of 2.300 to 3.000 indicating a highly probable identification at the species level, a score of 2.000 to 2.299 indicating secure genus identification with probable species identification, a score of 1.700 to 1.999 suggesting probable identification at the genus level, and scores below 1700 considered unreliable for identification [42].

### 2.4. Statistical Analysis

All measurements and analyses were conducted in triplicate. The mean and standard deviation (SD) for the microorganism counts were calculated using Microsoft Excel software 2311. One-way ANOVA (main factor: treatment) was performed using Prism 8.0.1 (GraphPad Software, San Diego, CA, USA), followed by Tukey’s test at α = 0.05 for post hoc analysis. Data processing was carried out using SAS^®^ version 8 software.

## 3. Results 

### 3.1. Microbiological Quality of Ewe’s Cheese

The effect of different treatments on the selected microbial populations at different sampling times is showed in Figure 1, Figure 2, Figure 3 and Figure 4, while that measured as the end of the observation period (15 days of storage) in comparison with time zero and controls.

Without treatment (C 15 and CV 15 samples), vacuum packaging significantly reduced the growth of TVC, CB, and MFF compared to air-stored sample.

Further, the spoilage of both undesirable bacterial strains (CB) as well as observed fungi (MFF) appeared to be significantly reduced in comparison with all the controls (C 15, CV 15, CSO 15) when both EOs and corresponding dried herb were used for treatment. On the other hand, LAB succeeded in their development in all treated cheeses.

All in all, regardless of the microbial population observed, the effectiveness showed by dried herbs appears substantially comparable with that observed by using the corresponding essential oil.

Among treatments, while oregano EO was the most effective against MFF, dried oregano showed the strongest inhibitory effect against both TVC and CB, together with the lowest activity against LAB growth.

In further detail, starting from an initial average value of 3.68 ± 0.02 log CFU/g, after 15 days of storage, the percentage difference between the average number of TVC in the control and the samples treated with DO was 21% (Figure 1). Regarding CB, the result was visibly already after the 5th day of storage, with the average concentration ranging from 2.27 ± 0.02 log CFU/g in the samples treated with REO to 2.61 ± 0.01 log CFU/g in the control group, while after 15th day of storage the highest difference (~37%) was observed between the control and cheese treated with DR (Figure 2). With respect to MFF, from the initial concentration of 1.55 ± 0.02 log CFU/g, OEO treatment gave the best result, which was already visible after five days of storage, with an increase of ~65% over the initial value compared to the control which showed a 123% increase (Figure 3). The average quantity of LAB during time varied according to the storage method used: 3.03 ± 0.02 log CFU/g for samples treated with DR to 3.46 ± 0.02 log CFU/g for samples treated with DO after the fifth day, 3.42 ± 0.02 log CFU/g for samples treated with REO to 3.97 ± 0.02 log CFU/g for samples treated with DO after the tenth day, and 4.10 ± 0.01 log CFU/g for samples treated with DR to 4.49 ± 0.01 log CFU/g in the samples treated with DO after the fifteenth day of storage (Figure 4).

The results are confirmed by the microorganism growth rate during time. The rate constant (k) was indeed used as a quantitative parameter to test whether the treatments studied could differently slow down or promote the growth of microorganisms during the observation period (15 days). The k values (Table 2) were calculated considering an order one kinetic according to the following equation:d[M]_t=t_/dt = k_m,T_ × [M]_t=t_(1)
where km, T is the kinetic constant and [M]_t=t_ is the microbic concentration at time t = t.

After integration, the following equation is obtained:[M]_t=t_ = [M]_t=0_ × e^km,T×t^(2)
where [M]_t=0_ is the microbic concentration at time t = 0.

In a logarithmic scale, the equation obtained is linear (Figure 1, Figure 2, Figure 3 and Figure 4):ln [M]_t=t_ = ln [M]_t=0_ + k_m,T_ × t(3)

A strong correlation between theoretical and experimental data (R^2^) was also generally observed.

According to what has been reported by the authors mentioned above, our results showed that, as desired, all the essential oils tested did not completely counteract the growth of LAB. Furthermore, regarding the development of LAB, dried herbs gave comparable or in some cases better results than the corresponding EOs.

### 3.2. Identification of Isolated Microorganisms

In the present study, 7 families, 10 genera, and 11 species were identified. The most frequently isolated microorganisms belong to the family Enterobacteriaceae, followed by Lactobacillaceae, Streptococacceae, Pseudomonadaceae, Saccharomycetaceae, Dipodascaceae, and Yersiniaceae. The most represented species was *Citrobacter baraakii* (21%), followed by *Hafnia alvei* and *Lacticaseibacillus paracasei* (16%) (Figure 5).

The bacteria isolated and the number of isolates is shown in Table 3. A total of 317 isolates were identified in the study. After 15 days of storage, different microbial distributions were highlighted as a function of treatment, with the highest microbial presence in the control samples.

*Citrobacter braakii*, *Hafnia alvei*, *Lacticaseibacillus paracasei*, *Lactococcus lactis*, and *Leuconostoc mesenteroides* were isolated regardless of the treatment. *Citrobacter freundi* was isolated from all control groups and from samples treated with oregano essential oil. *Ewingella americana* and *Pseudomonas fragi* were isolated only from the control samples. Yeats *Debaromyces hansenii, Klueyveromyces lactis*, and *Yarrowia lipolytica* were also isolated from cheese.

## 4. Discussion

Natural ingredients are recognized for their antioxidant properties, contributing to the improved microbiological quality of food [43]. In a study by Sadeghi et al. [44], the antimicrobial efficacy of mint essential oil (EO) at a concentration of 0.03% was assessed against *Listeria monocytogenes* in Iranian white cheese. In the control group, *Listeria monocytogenes* exhibited growth within 7 days, slowing down over the subsequent 60 days of storage. Contrastingly, all samples treated with mint essential oil displayed a logarithmic reduction in bacterial count, with the maximum growth observed for 14 days, revealing a significant difference (*p* < 0.05) compared to the control samples. Similarly, in our experiment, the counts of total viable bacteria (TVC) and coliform bacteria (CB) were significantly lower (*p* < 0.05) in the samples treated with essential oil and dried herbs compared to the control groups throughout the entire storage period. 

In our study, we investigated the impact of 1% oregano, thyme, and rosemary essential oils, as well as dried oregano, thyme, and rosemary, on the microbiological quality of ewe’s cheese during storage. Dried oregano exhibited the most potent inhibitory effect against TVC, while it was also the most effective against CB. Oregano essential oil demonstrated the highest efficacy against yeasts and fungi.

Govaris et al. [45] investigated the antibacterial properties of 1% and 2% oregano essential oil (EO) and 1 % thyme essential oil against *Listeria monocytogenes* and *Escherichia coli* O157:H7 in feta cheese under modified atmospheric conditions (50% CO_2_ and 50% N_2_) at 4 °C. Their findings indicated that *Listeria monocytogenes* and *Escherichia coli* O157:H7 remained viable in samples with 1 % oregano and thyme essential oils for 18 and 22 days of storage, respectively, while these bacteria persisted for 14 and 16 days in samples treated with 2% oregano essential oil. In our study, 1% oregano and 1% thyme oil were also employed, albeit in combination with vacuum packaging.

In our study, we assessed the microbiological quality of cheese treated with essential oils (EO) and dried herbs over a 15-day storage period. Drawing parallels, Kavas et al. [46] investigated the impact of thyme and clove essential oils (1.5% *v*/*v*) incorporated into an edible film based on whey protein isolate in Kashar cheese, focusing on the growth of Escherichia coli over a 60-day storage period. The study revealed that thyme essential oil exhibited superior antimicrobial activity compared to clove essential oil. In cheese samples treated with thyme and clove essential oils, the *Escherichia coli* count was 5.86 and 6.12 log CFU/g on day 0, respectively. By the 60th day of storage, the number of *Escherichia coli* had decreased by 2.25–4.49 and 1.57–4.91 log CFU/g, respectively. In our investigation, the count of coliform bacteria increased during storage, although the growth of microorganisms was attenuated in samples treated with essential oils and dried herbs compared to control samples.

Moro et al. [20] conveyed that lactic acid bacteria (LAB) found in cheese exhibit resilience to the effects of plant extracts and essential oils (EOs) at concentrations effective against the growth of pathogenic bacteria. Specifically, they observed that rosemary EO inhibited the growth of *Clostridium tyrobutyricum* in sheep cheese but did not decrease the LAB population. Our study aligns with these findings, indicating a limited impact of essential oils and dried herbs against LAB. In a similar vein, Marcial et al. [23] found that the addition of 200 mg/kg oregano EO to cheese had no discernible effect on the growth of *Lactococcus lactis* subsp. *lactis*, *Lactobacillus delbrueckii* subsp. *bulgaricus*, and *Streptococcus thermophilus* compared to control samples. Conversely, De Carvalho et al. [21] reported a contrasting outcome, noting that essential oils have the capacity to inhibit LAB, resulting in an elevation of cheese pH. They specifically observed that a concentration of 2.5 μL/mL of thyme EO led to a reduction in the number of starter cultures, composed of *Lactococcus lactis* subsp. *cremoris* and *Lactococcus lactis* subsp. *lactis*, to approximately 4 log CFU/mL.

Zandona et al. [47] explored the impact of incorporating organic hemp seed powder in conjunction with vacuum packaging (VP) and modified atmosphere conditions (MAP) on the microbiological quality of traditional Croatian whey cheese over a 21-day storage period. Contrary to their findings, our results demonstrated that the addition of hemp seed powder did not positively influence shelf life. In our study, vacuum packaging exhibited a beneficial effect by reducing the growth of total viable bacteria (TVC), coliform bacteria (CB), yeasts, and fungi when compared to aerobically packaged samples. Licon [24] investigated the antimicrobial efficacy of citronella and thyme essential oils against *Escherichia coli*, *Clostridium tyrobutyricum*, and *Penicillium verrucosum* in sheep’s cheese. Their results indicated that thyme essential oil reduced the numbers of *Clostridium tyrobutyricum* and completely inhibited the growth of *Penicillium verrucosum* without impacting the natural flora in the cheese. However, citronella essential oil exhibited an inhibitory effect against starter cultures of lactic acid bacteria, rendering it unsuitable for cheeses. In our study, essential oils and dried herbs did not significantly affect the number of lactic acid bacteria (LAB). Muñoz-Tébar et al. [48] delved into the influence of adding oil extracted from the seeds of *Salvia hispanica* L. (chia) to sheep’s cheese on the occurrence of selected groups of microorganisms. Their findings showed no significant differences between control cheeses and those enriched with chia oil concerning total bacteria and the two groups of lactic acid bacteria. De Souza et al. [22] investigated the inhibitory effects of oregano essential oil at different concentrations (0.6%, 1.25%, 2.5%, and 5%) on *Staphylococcus aureus*, *Listeria monocytogenes*, and *Lactococcus lactis*. While our study aligns with their observations regarding the limited impact on LAB, they reported that oregano essential oil with a concentration of 1.25% caused a reduction in the numbers of *L. monocytogenes* and *S. aureus* after 24 h, with this concentration also inhibiting LAB.

Traditional or local farmhouse lump cheeses crafted from ewe’s milk exhibit a diverse microbiota, with Gram-positive bacterial species naturally present in fresh ewe’s milk lump cheeses. Lactic acid bacteria (LAB) typically constitute the predominant population in raw milk [49]. In lump cheeses derived from ewe’s milk, the microbiome often includes taxa from LAB, such as *Lactococcus*, *Streptococcus*, and/or *Enterococcus* [50]. Our study identified *Lacticaseibacillus paracasei*, *Lactococcus lactis*, and *Leuconostoc mesenteroides* as the most frequently isolated species from LAB, all of which play essential roles in the cheese ripening process.

The microbiota of lump sheep cheeses derived from sheep’s milk displays variability, with Gram-positive bacterial genera associated with their natural occurrence. Predominantly, lactic acid bacteria (LAB) dominate the microbial composition [49]. In A study by Kačániová et al. [51] exploring the microbiota diversity in Slovak summer sheep cheese, Bryndza, identified *Yarrowia lipolitica*, *Lacticaseibacillus paracasei* subsp. *paracasei*, and *Dipodascus geotrichum* as the most frequently isolated microbial species. The Lactobacillaceae family was the most numerous in “Bryndza” cheeses. Additionally, Lauková et al. [50] investigated the microbiota of ewe’s milk lump cheeses, finding that the most represented genus was *Streptococcus* (belonging to the family Streptococcaceae), followed by *Lactococcus*, also within the family Streptococcaceae. 

Additionally, the elevated levels of protein, lipid, minerals, vitamins, and moisture in cheese products create a conducive environment for the proliferation of pathogenic and spoilage bacteria, leading to a decline in product quality or rendering them unsuitable for human consumption [52]. While susceptibility to specific microorganisms varies among cheese types, bacterial, yeast, and fungal activity are common contributors to cheese deterioration [3]. In our study, the most frequently isolated bacteria were *Citrobacter braakii*, and the predominant family in ewe’s cheese was Enterobacteriaceae. Sangoyomi et al. [53] identified eight bacterial genera and yeasts, with *Lactobacillus* being the most commonly occurring LAB genus. The Enterobacteria group comprised *Escherichia coli*, *Klebsiella*, and *Enterobacter*. Psychrotrophic pseudomonads, capable of thriving at low temperatures, may appear in cheeses, producing enzymes that break down proteins. However, the presence of harmful bacteria can also be attributed to inadequate hygiene practices in cheese production [54]. In our study, *Pseudomomas fragi* was consistently isolated from all control groups, irrespective of storage conditions, whereas Pseudomonas bacteria were not detected in samples treated with essential oils and dried herbs.

Increased levels of protein and fat in the diet could potentially interact with components of essential oils differently than under simulated conditions [13]. Moreover, the richer nutritional composition of cheese provides an optimal environment for bacteria to repair their damaged cells [13,55].

## 5. Conclusions

Modern consumers increasingly seek food that is free from synthetic preservatives. Consequently, there is a growing exploration of natural substances with antioxidant and antimicrobial properties to meet this demand. Our study reveals that dried oregano, thyme, and rosemary, along with their respective 1% essential oils, can effectively inhibit the growth of microorganisms in fresh sheep’s milk cheese when combined with vacuum packaging. The essential oils and dried herbs utilized in our experiment demonstrated a significant reduction in the counts of total viable bacteria (TVC), coliform bacteria (CB), and microscopic filamentous fungi (MFF) over a 15-day storage period, without notably impacting the count of lactic acid bacteria (LAB). While all the natural substances exhibited inhibitory effects against the studied microorganisms, the most potent inhibitory effects were observed with oregano essential oil and dried oregano. Importantly, under our experimental conditions, dried herbs exhibited comparable effectiveness to their corresponding essential oils, affirming their potential as cost-effective natural preservatives to extend the shelf life of lump cheese.

## Figures and Tables

**Figure 1 foods-12-04487-f001:**
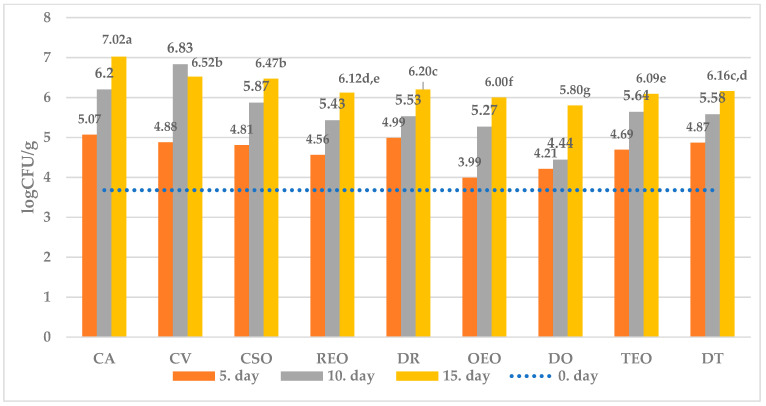
Average numbers (log CFU/g) of TVCs in samples of ewe’s cheese during 15 days of storage at 4 °C. CA—aerobically packaged control samples; CV—vacuum packaged control samples; CSO—vacuum packaged control samples treated with sunflower oil; REO—vacuum packaged samples treated with 1% rosemary EO; TEO—vacuum packaged samples treated with 1% thyme EO, OEO—vacuum packaged samples treated with 1% oregano EO, DR—vacuum packaged samples treated with dried rosemary, DT—vacuum packaged samples treated with dried thyme, DO—vacuum packaged samples treated with dried oregano. *p*-value < 0.001; in the same row different letters indicate statistically significant differences.

**Figure 2 foods-12-04487-f002:**
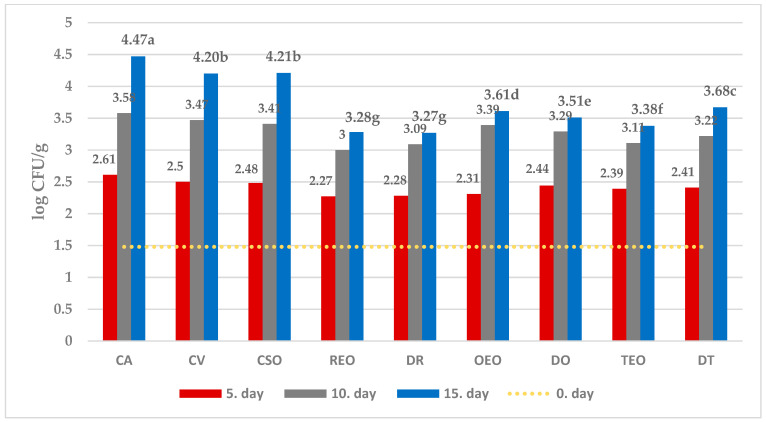
Average numbers (log CFU/g) of CB in samples of ewe’s cheese during 15 days of storage at 4 °C. CA—aerobically packaged control samples; CV—vacuum packaged control samples; CSO—vacuum packaged control samples treated with sunflower oil; REO—vacuum packaged samples treated with 1% rosemary EO; TEO—vacuum packaged samples treated with 1% thyme EO, OEO—vacuum packaged samples treated with 1% oregano EO, DR—vacuum packaged samples treated with dried rosemary, DT—vacuum packaged samples treated with dried thyme, DO—vacuum packaged samples treated with dried oregano. *p*-value < 0.001; in the same row different letters indicate statistically significant differences.

**Figure 3 foods-12-04487-f003:**
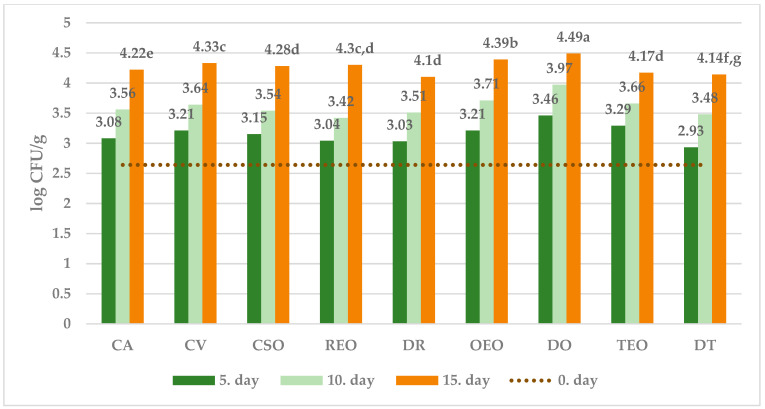
Average numbers (log CFU/g) of MFF in samples of ewe’s cheese during 15 days of storage at 4 °C. CA—aerobically packaged control samples; CV—vacuum packaged control samples; CSO—vacuum packaged control samples treated with sunflower oil; REO—vacuum packaged samples treated with 1% rosemary EO; TEO—vacuum packaged samples treated with 1% thyme EO, OEO—vacuum packaged samples treated with 1% oregano EO, DR—vacuum packaged samples treated with dried rosemary, DT—vacuum packaged samples treated with dried thyme, DO—vacuum packaged samples treated with dried oregano. *p*-value < 0.001; in the same row different letters indicate statistically significant differences.

**Figure 4 foods-12-04487-f004:**
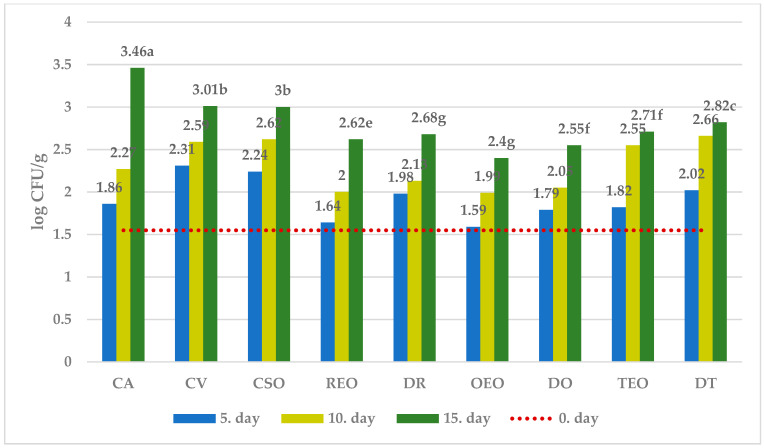
Average numbers (log CFU/g) of LAB in samples of ewe’s cheese during 15 days of storage at 4 °C. CA—aerobically packaged control samples; CV—vacuum packaged control samples; CSO—vacuum packaged control samples treated with sunflower oil; REO—vacuum packaged samples treated with 1% rosemary EO; TEO—vacuum packaged samples treated with 1% thyme EO, OEO—vacuum packaged samples treated with 1% oregano EO, DR—vacuum packaged samples treated with dried rosemary, DT—vacuum packaged samples treated with dried thyme, DO—vacuum packaged samples treated with dried oregano. *p*-value < 0.001; in the same row different letters indicate statistically significant differences.

**Figure 5 foods-12-04487-f005:**
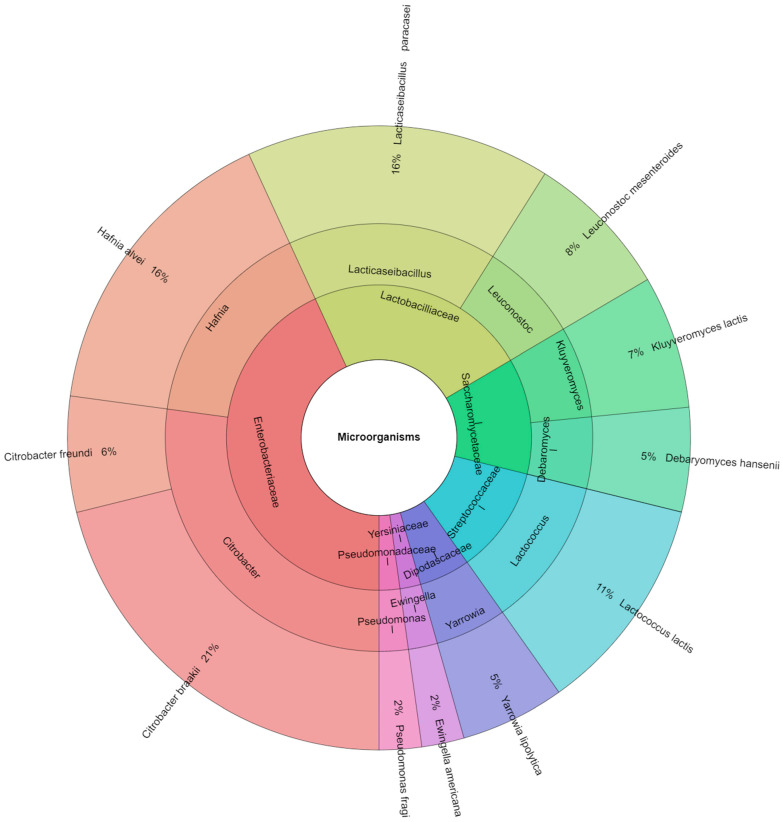
Krona chart: isolated species, genera, and families of bacteria in the ewe’s cheese during observation period.

**Table 1 foods-12-04487-t001:** Protocol for preparation of ewe’s cheese sample.

Control group	CA	samples of ewe’s cheese packaged under aerobic conditions and stored at temperature 4 °C
Control group with vacuum packaging	CV	samples of ewe’s cheese, were vacuum-sealed in polyethylene bags and stored at a temperature of 4 °C
Control group with sunflower oil	CSO	samples of ewe’s cheese treated with sunflower oil, were vacuum-sealed in polyethylene bags and stored at a temperature of 4 °C
Samples with 1% rosemary essential oil	REO	samples of ewe’s cheese treated with 1% REO, were vacuum-sealed in polyethylene bags and stored at a temperature of 4 °C
Samples with 1% thyme essential oil	TEO	samples of ewe’s cheese treated with 1% TEO, were vacuum-sealed in polyethylene bags and stored at a temperature of 4 °C
Samples with 1% oregano essential oil	OEO	samples of ewe’s cheese treated with 1% OEO, were vacuum-sealed in polyethylene bags and stored at a temperature of 4 °C
Samples with dried rosemary	DR	samples of ewe’s cheese treated with 1% DR, were vacuum-sealed in polyethylene bags and stored at a temperature of 4 °C
Samples with dried thyme	DT	samples of ewe’s cheese treated with 1% DT, were vacuum-sealed in polyethylene bags and stored at a temperature of 4 °C
Samples with dried oregano	DO	samples of ewe’s cheese treated with 1% DO, were vacuum-sealed in polyethylene bags and stored at a temperature of 4 °C

**Table 2 foods-12-04487-t002:** Kinetic constants related to the time evolution of the microbiological concentration evaluated (TVC = total viable counts; CB = coliform bacteria; LAB = lactic acid bacteria; MFF = microscopic filamentous fungi).

Sample	TVC (k ± c.i.) × 10^−1^	R^2^	CB (k ± c.i.) × 10^−1^ [day^−1^]	R^2^	LAB (k ± c.i.) × 10^−1^ [day^−1^]	R^2^	MFF (k ± c.i.) × 10^−1^ [day^−1^]	R^2^
	[day^−1^]							
C	2.35 ± 0.01	0.987	1.87 ± 0.01	0.999	1.00 ± 0.03	0.990	1.09 ± 0.01	0.923
CV	2.00 ± 0.01	0.986	1.71 ± 0.02	0.997	1.09 ± 0.02	0.992	1.22 ± 0.17	0.780
CSO	1.96 ± 0.02	0.988	1.69 ± 0.02	0.999	1.03 ± 0.03	0.982	1.02 ± 0.02	0.974
REO	1.67 ± 0.01	0.997	1.15 ± 0.01	0.973	0.99 ± 0.02	0.952	0.60 ± 0.02	0.899
DR	1.80 ± 0.02	0.958	1.17 ± 0.02	0.952	0.93 ± 0.03	0.991	0.71 ± 0.02	0.957
TEO	1.74 ± 0.01	0.974	1.24 ± 0.02	0.966	1.04 ± 0.02	0.988	0.82 ± 0.01	0.939
DT	1.78 ± 0.01	0.971	1.40 ± 0.03	0.986	0.93 ± 0.02	0.972	0.92 ± 0.01	0.957
OEO	1.49 ± 0.02	0.955	1.41 ± 0.02	0.949	1.16 ± 0.03	0.996	0.50 ± 0.02	0.915
DO	1.20 ± 0.02	0.886	1.34 ± 0.01	0.955	1.29 ± 0.01	0.985	0.60 ± 0.02	0.966

**Table 3 foods-12-04487-t003:** Isolated bacteria from samples of ewe’s cheese after 15 days of storage at 4 °C.

Microorganisms	Sample									
	CA	CV	CSO	REO	TEO	OEO	DR	DT	DO	Total
*Citrobacter braakii*	19	15	14	4	5	3	3	2	2	67
*Citrobacter freundi*	9	6	3	-	-	1	-	-	-	19
*Debaryomyces hansenii*	5	5	3		1	2	1	-	-	17
*Hafnia alvei*	17	12	11	3	2	1	2	2	1	51
*Ewingella americana*	4	2	1	-	-	-	-	-	-	7
*Kluyveromyces lactis*	3	5	3	1	1	3	2	3	1	22
*Pseudomonas fragi*	4	2	1	-	-	-	-	-	-	7
*Lacticaseibacillus paracasei*	2	8	5	4	5	8	4	5	9	50
*Lactococcus lactis*	5	3	7	5	3	5	4	2	2	36
*Leuconostoc mesenteroides*	1	5	4	2	1	1	3	2	5	24
*Yarrowia lipolytica*	2	3	2	3	-	3	2	1	1	17
Total	71	66	54	22	18	27	21	17	21	317

CA—aerobically packaged control samples; CV—vacuum packaged control samples; CSO—vacuum packaged control samples treated with sunflower oil; REO—vacuum packaged samples treated with 1% rosemary EO; TEO—vacuum packaged samples treated with 1% thyme EO, OEO—vacuum packaged samples treated with 1% oregano EO, DR—vacuum packaged samples treated with dried rosemary, DT—vacuum packaged samples treated with dried thyme, DO—vacuum packaged samples treated with dried oregano.

## Data Availability

Data are contained within the article.

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
