# Peer review of "Dried Herbs as an Easy-to-Use and Cost-Effective Alternative to Essential Oils to Extend the Shelf Life of Sheep Lump Cheese"

_foods, 2023, doi:10.3390/foods12244487_

Round 1

Reviewer 1 Report

Comments and Suggestions for Authors

The manuscript submitted for review concerns the influence of dried herbs (rosemary, thyme and oregano) on the microbiological quality of sheep lump cheese. The authors compared the obtained results with essential oils (EOs) corresponding to the mentioned herbs. It should be emphasized that the authors planned the experiments appropriately and performed many experimental variants. However, it seems to me that the authors formulated the purpose of the research somewhat incorrectly (last paragraph in the Introduction). The goal contains too many Methodology elements. Please establish the research goal precisely and concisely. Moreover, the presentation of the results in Figures 1-4 is illegible. Maybe it would be better to place controls, herbs and EOs on the X axis, then changes in the number of bacteria over time would be more clear for each "treatment" used (of course, a separate graph for each group of bacteria). Please also correct "typos" - e.g. line 272: it should be "represented species" instead of "represented specie".

Author Response

Reviewer #1

The manuscript submitted for review concerns the influence of dried herbs (rosemary, thyme and oregano) on the microbiological quality of sheep lump cheese. The authors compared the obtained results with essential oils (EOs) corresponding to the mentioned herbs. It should be emphasized that the authors planned the experiments appropriately and performed many experimental variants.

We would like to thank the Reviewer for the time devoted for constructive and important comments to improve our paper.

 Point 1: However, it seems to me that the authors formulated the purpose of the research somewhat incorrectly (last paragraph in the Introduction). The goal contains too many Methodology elements. Please establish the research goal precisely and concisely.

Response: It was corrected.

Point 2: Moreover, the presentation of the results in Figures 1-4 is illegible. Maybe it would be better to place controls, herbs and EOs on the X axis, then changes in the number of bacteria over time would be more clear for each "treatment" used (of course, a separate graph for each group of bacteria).

Response: It was changed by the instruction.

Point 3: Please also correct "typos" - e.g. line 272: it should be "represented species" instead of "represented specie".

Response: It was corrected.

Reviewer 2 Report

Comments and Suggestions for Authors

I suggest:

- using the complete nomenclature of the plants, at least once in the text (i.e. in the Introduction or Material and Methods paragraph), such as Rosmarinus officinalis L., Origanum vulgare L., Thymus vulgaris L..

Also adding these plants belong to the Lamiaceae family of which several species show high pesticidal activity (and hese references: Ebadollahi, A.; Ziaee, M.; Palla, F. Essential oils extracted from different species of the Lamiaceae plant family as prospective bioagents against several detrimental pests. Molecules 2020, 25, 1556; Stankovic, M. Lamiaceae Species: Biology, Ecology and Practical Uses; MDPI: Basel, Switerland, 2020;).

-Figures 1 to 4, the improvement of graphic structure, does not allow quick and clear reading of the results

Author Response

Reviewer #2

Point 1: I suggest:

- using the complete nomenclature of the plants, at least once in the text (i.e. in the Introduction or Material and Methods paragraph), such as Rosmarinus officinalis L., Origanum vulgare L., Thymus vulgaris L.

Also adding these plants belong to the Lamiaceae family of which several species show high pesticidal activity (and these references: Ebadollahi, A.; Ziaee, M.; Palla, F. Essential oils extracted from different species of the Lamiaceae plant family as prospective bioagents against several detrimental pests. Molecules 2020, 25, 1556; Stankovic, M. Lamiaceae Species: Biology, Ecology and Practical Uses; MDPI: Basel, Switerland, 2020;).

Response: We would like to thank the Reviewer for the time devoted for constructive and important comments to improve our paper. The Latin names, family Lamiaceae, were mentioned in the manuscript in part Introduction. We also mentioned in manuscript pesticidal activity which references were mentioned in the reviewer opinion.

Point 2: Figures 1 to 4, the improvement of graphic structure, does not allow quick and clear reading of the results.

Response: Figures presentations were changed.

Reviewer 3 Report

Comments and Suggestions for Authors

Dear authors! Your article is covering interesting topic about improvement of stability of food by herbs. 

Please find remarks about article:

1.    Line 82 - total LAB is not always correct parameter, specific LAB strains are necessary for cheese. How you can prove that in your research are LABS necessary for exact cheese type? May be it is undesirable microflora?

2.    Line 115 - Please check term - efficacy

3.     LINE138 - 5g of sample - why so small sample? It is not representative. Please specify sample size and number of packaging’s tested each step (0, 5 10, 15 days)

4.     Line 140 - how herbs were pre-treated - grounded, powdered? Participle size? It could affect extraction of antimicrobials in media and their possible activity.

5.     Line 140 - how practically you soaked 5 g with 100 microliters? How was distribution of EO and herbs with oil in cheese sample?

6.    Line 146-line 158 - please add standards for microbiological methods

7.       Materials and methods - you added herbs in oil and than applied to cheese. Are these antimicrobials fat soluble and could extract in oil and further work in cheese?

8.       Line 147 in previous page you wrote that sample is 5 g and here that sample for analyses were cut in 5 g pieces?

9.    How it is possible to dissolve 1 g of herbs in 100 microliters of oil?

10.  Figure 1. Please format it according to requirements. It is necessary to use another way of presentation where it is possible to see variation of data (data of different samples are too close to each other).

11.  Results - did you evaluated microorganisms on the surface of herbs? Very often herbs are rich in microorganisms.

12.  Line 178 - how many parallel packaging’s you took for analyses each time? It was written that in total 144 samples were prepared (each 5 g?)?

13.  Table 2 - I am wondering about standard deviations and variation between repetitions of microbial analyses because you show significant differences between 6.2 and 6.12 Please explain in detail how you did it.

14.   Figure 4 and also others are not very clear. All lines are close to each other. Please find some other way of presentation of data.

15.   In table 2 - please add units of values presented in table

16.  Discussion - please emphasis desirable LAB for this cheese and desirable LAB.

17.  References - please format numbering of references

18.  About pathogenic microorganisms - why they are present in your sample and increase so fast? To confirm that herbs work as antimicrobials, than it is necessary to contaminate with pathogens and evaluate their development in media of herbs.

19.  Line 363 - it was necessary to add it in introduction - characterization of microorganisms that need to be inhibited.

20.  Line 377 Pseudomonads produce enzymes and break down proteins into products - please specify what kind of products?

21.   Line 388 - did you confirmed that it extend shelf life?

 Generally – description of design should be improved significantly, and presentation of results and data treatment.

Author Response

Reviewer #3

Dear authors! Your article is covering interesting topic about improvement of stability of food by herbs. 

Please find remarks about article:

We would like to thank the Reviewer for the time devoted for constructive and important comments to improve our paper.

Point 1:    Line 82 - total LAB is not always correct parameter, specific LAB strains are necessary for cheese. How you can prove that in your research are LABS necessary for exact cheese type? May be it is undesirable microflora?

Response: In different study were found that essential oil can decreased the number of LAB in 1 to 4 log CFU/mL. Some study also shows that essential oil doesn’t have impact to pathogenic bacteria and Lactic acid bacteria. We are understand that LAB are necessary for cheese. In our study were evaluated different group of microorganisms and from our knowledge were LAB also evaluated, if additive on basis of essential oils and herbs can influent number of microorganisms. The isolated species from our study is normal microbiota of unpasteurized ewe’s cheese. Also exist study that LAB can be hurdle for spoilage bacteria in cheese. And for these different questions were LAB evaluated in our study.

Here are some examples of microbiota of cheese after application of EO and how LAB can eliminate spoilage bacteria:

Khorshidian, N., Yousefi, M., Khanniri, E., & Mortazavian, A. M. (2018). Potential application of essential oils as antimicrobial preservatives in cheese. Innovative Food Science & Emerging Technologies, 45, 62–72. https://doi.org/10.1016/j.ifset.2017.09.020.

Settanni, L., Gaglio, R., Guarcello, R., Francesca, N., Carpino, S., Sannino, C., & Todaro, M. (2013). Selected lactic acid bacteria as a hurdle to the microbial spoilage of cheese: Application on a traditional raw ewes’ milk cheese. International Dairy Journal, 32(2), 126–132. https://doi.org/10.1016/j.idairyj.2013.04.010.

Point 2:    Line 115 - Please check term – efficacy.

Response: It was changed.

Point 3:     LINE138 - 5g of sample - why so small sample? It is not representative. Please specify sample size and number of packaging’s tested each step (0, 5 10, 15 days).

Response: The number of grams were corrected.

Point 4:     Line 140 - how herbs were pre-treated - grounded, powdered? Participle size? It could affect extraction of antimicrobials in media and their possible activity.

Response: The samples were grounded by producer. The manufacturer states that the herbs were dried and ground, we did not measure the size, approximately 2-3x2-3 mm. In our experiment was mixed herbs used, normally for antimicrobial activity of herbs were doing ethanolic extracts. In our study we used mixing with sunflower oil and other procedure were added to material and methods.

Point 5:  Line 140 - how practically you soaked 5 g with 100 microliters? How was distribution of EO and herbs with oil in cheese sample?

Response: It was described in material and methods. Microlitres was mistakes and 1 g of herbs were mixed with sunflower oil.

Point 6: Line 146-line 158 - please add standards for microbiological methods.

Response: It was added.

Point 7: Materials and methods - you added herbs in oil and than applied to cheese. Are these antimicrobials fat soluble and could extract in oil and further work in cheese?

Response: In our work, we were based on the knowledge that plant essential oils are very difficult to dissolve in water, and therefore we also used oil for herbs to make the results comparable. Some study they state that they also dissolve in fats, we used oil in our work when applying it to the mixing with the cheese. There is no evidence that herbs dissolved in oil have a better antimicrobial effect in food.

Point 8: Line 147 in previous page you wrote that sample is 5 g and here that sample for analyses were cut in 5 g pieces?

Response: It was corrected.

Point 9:    How it is possible to dissolve 1 g of herbs in 100 microliters of oil?

Response: It was mistakes. In material and method was done 1 g in 100 mL.

Point 10:  Figure 1. Please format it according to requirements. It is necessary to use another way of presentation where it is possible to see variation of data (data of different samples are too close to each other).

Response: Figures presentations were changed.

Point 11:  Results - did you evaluated microorganisms on the surface of herbs? Very often herbs are rich in microorganisms.

Response: Thes sentence was added to material and methods. Testing additives for microbiology is a necessity for the food industry. The work used herbs that are found in commercial networks and must meet microbiological requirements. It was not mentioned in the text of the article because this information results from generally known regulations.

Point 12:  Line 178 - how many parallel packaging’s you took for analyses each time? It was written that in total 144 samples were prepared (each 5 g?)?

Response: A total of 144 cheese samples were analyzed in the work. The samples were tested during storage 4 times (0. day, 5. day, 10. day and 15. day) it follows that 4 samples from each variant were examined for microbiology in three repetitions.

Point 13:  Table 2 - I am wondering about standard deviations and variation between repetitions of microbial analyses because you show significant differences between 6.2 and 6.12 Please explain in detail how you did it.

Response: One-way ANOVA (main factor: treatment) was performed, followed by Tukey's test at α = 0.05. For example, about the results the Reviewer was asking about, the standard deviation was very low, namely 6.12±0.03 and 6.20±0.01. We have added standard deviation in the table in order to better show the differences. Moreover, we realized that a couple of values were put in wrong rows. We apologize for the error, we have corrected the table.

Point 14:   Figure 4 and also others are not very clear. All lines are close to each other. Please find some other way of presentation of data.

Response: Figures presentations were changed.

Point 15:   In table 2 - please add units of values presented in table.

Response: It was added.

Point 16:  Discussion - please emphasis desirable LAB for this cheese and desirable LAB.

Response: It was added.

Point 17:  References - please format numbering of references.

Response: It was changed.

Point 18:  About pathogenic microorganisms - why they are present in your sample and increase so fast? To confirm that herbs work as antimicrobials, then it is necessary to contaminate with pathogens and evaluate their development in media of herbs.

Response: Dear reviewer the number of increases during storage. In our work, no food pathogens were detected, only representatives of coliform bacteria and Pseudomonas fragi, and only in the control group of samples. It goes without saying that the number of microorganisms increases during food storage. In our work, we proved that the addition of essential oils and herbs slowed down the growth of spoilage bacteria. In different works, we also deal with the antimicrobial activity of plant essential oils and herbs in relation to meat, with pathogenic bacteria. In our experiment, we are dealing with cheese without heat treatment, and the goal of our work was to prove that the addition of natural preservatives has potential. There is a large number of data reporting the antimicrobial potential of herbal extracts. Their effectiveness is recorded in a large number of works. Of course, many studies still need to be done to prove their antimicrobial properties on a food model, because it depends on the product and the speed of its influence by the effect of spoilage bacteria.

Point 19:  Line 363 - it was necessary to add it in introduction - characterization of microorganisms that need to be inhibited.

Response: In the discussion part some information related to bacteria in cheese and essential oils were mentioned.

Point 20:  Line 377 Pseudomonads produce enzymes and break down proteins into products - please specify what kind of products?

Response: Proteases catalyze the hydrolysis of peptide bonds of protein substrates. They catalyze the hydrolysis of proteins into peptides and lipopeptides into amino acids. As examples of products are cheese, milk, meat, eggs, fish products.

Point 21.   Line 388 - did you confirmed that it extends shelf life?

Response: Unpasteurized cheese was used in the work, and therefore it is more likely that the number of microorganisms increases over time. Our results showed that due to the effect of plant essential oils and herbs, the number of microorganisms was lower compared to the control. These findings point to the possibility of using plants as natural substances to extend the shelf life of food.

Point 22: Generally – description of design should be improved significantly, and presentation of results and data treatment.

Response: Description were improved results were corrected, introduction, material and methods and discussion thoroughly rewritten.

Round 2

Reviewer 3 Report

Comments and Suggestions for Authors

Dear authors! Thanks for your work, improvements are significant, but still I have few comments.

1) Line 453 - were microbiologically tested prior to application on cheese samples, with no bacterial presence detected – Please specify how you did it and how is about moulds and yeast that further can spoil cheese?

2) Table 2 shows the same data presented in graph – I suggest to add letters in graph – than we can see all changes . You can add  graph comparing days and zero day add as line, other wise you have the same result 9 times in one graph. Templeate in attached file.

3) I am still wondering of so small standard deviation between 4 repetition samples – it is around 1 % from mean (variation coefficient). In other scientists work we often see it at least 3%, 5% and even more for more variable matrixes. How you explain so similar results, because it affects all your conclusion, that is why I ask you this question second time.

Best regards

Author Response

We would like to thank the Reviewer for the time devoted for constructive and important comments to improve our paper.

Point 1: Line 453 - were microbiologically tested prior to application on cheese samples, with no bacterial presence detected – Please specify how you did it and how is about moulds and yeast that further can spoil cheese?

Response: The sentence Line 558-559: The dried herbs (Rosmarinus officinalis, Origanum vulgare, Thymus vulgaris) were purchased from Mäspoma Ltd. (Zvolen, Slovakia) and were microbiologically tested before use on the cheese samples, and no bacteria were found to be present. This sentence was added to the material and method after the first round of revision. This sentence applied to the dry herbs, not to the cheese samples.

Microscopic fungi were not present in our samples, only basal fungi. From yeasts, Debaryomyces hansenii and Kluyveromyces lactis species were isolated. These yeasts are among the most abundant yeasts in several types of cheese. Yeasts are an important component of cheeses, especially artisan cheeses. In some types of cheese, yeasts are the main microbial group, at least for part of the ripening process. cheeses have a diverse range of yeast species. In our study, a typical Slovak fresh cheese with typical microflora such as yeasts and lactic acid bacteria was used. In the literature, all yeast species according to different studies of Slovak sheep cheese are reported.

In this manuscript is also mentioned species of yeasts from cheese: Quintana, Á. R., Perea, J. M., García-Béjar, B., Jiménez, L., Garzón, A., & Arias, R. (2020). Dominant Yeast Community in Raw Sheep’s Milk and Potential Transfers of Yeast Species in Relation to Farming Practices. Animals, 10(5), 906. https://doi.org/10.3390/ani10050906

Kačániová, M., Nagyová, Ľ., Štefániková, J., Felsöciová, S., Godočí­ková, L., Haščí­k, P., Horská, E., & Kunová, S. (2020). The characteristic of sheep cheese “Bryndza” from different regions of Slovakia based on microbiological quality. Potravinarstvo Slovak Journal of Food Sciences, 14, 69–75. https://doi.org/10.5219/1239

Kačániová, M., Terentjeva, M., Kunová, S., Haščík, P., Kowalczewski, P. Ł., & Štefániková, J. (2021). Diversity of microbiota in Slovak summer ewes’ cheese “Bryndza.” Open Life Sciences, 16(1), 277–286. https://doi.org/10.1515/biol-2021-0038

Point 2: Table 2 shows the same data presented in graph – I suggest to add letters in graph – than we can see all changes. You can add graph comparing days and zero day add as line, otherwise you have the same result 9 times in one graph. Template in attached file.

Response: Table 2 was removed and all information from tables is complete in new figures.

Point 3: I am still wondering about such a small standard deviation between 4 repetition samples – it is around 1 % from mean (variation coefficient). In other scientists' work we often see it at least 3%, 5% and even more for more variable matrices. How you explain so similar results, because it affects all your conclusion, that is why I ask you this question second time.

Response: We assume that the small deviations were due to the fact that the individual cheese samples were prepared from one lump of cheese and sliced for 144 pieces of cheese. Exist several works with products which mentioned small standard deviation. By the instruction of reviewer, we made new figures, where are statistically differences and Table 2 with SD were removed.